# Subcritical Transmission in the Early Stage of COVID-19 in Korea

**DOI:** 10.3390/ijerph18031265

**Published:** 2021-01-31

**Authors:** Yong Sul Won, Jong-Hoon Kim, Chi Young Ahn, Hyojung Lee

**Affiliations:** 1National Institute for Mathematical Sciences, 70, Yuseong-daero 1689 beon-gil, Yuseong-gu, Daejeon 34047, Korea; yong.won09@alumni.imperial.ac.uk (Y.S.W.); chiyoung@nims.re.kr (C.Y.A.); 2International Vaccine Institute, SNU Research Park, 1 Gwanak-ro, Gwanak-gu, Seoul 08826, Korea; JongHoon.Kim@ivi.int

**Keywords:** coronavirus, COVID-19, statistical model, serial interval, incubation period, pre-symptomatic transmission

## Abstract

While the coronavirus disease 2019 (COVID-19) outbreak has been ongoing in Korea since January 2020, there were limited transmissions during the early stages of the outbreak. In the present study, we aimed to provide a statistical characterization of COVID-19 transmissions that led to this small outbreak. We collated the individual data of the first 28 confirmed cases reported from 20 January to 10 February 2020. We estimated key epidemiological parameters such as reporting delay (i.e., time from symptom onset to confirmation), incubation period, and serial interval by fitting probability distributions to the data based on the maximum likelihood estimation. We also estimated the basic reproduction number ($R_{0})$ using the renewal equation, which allows for the transmissibility to differ between imported and locally transmitted cases. There were 16 imported and 12 locally transmitted cases, and secondary transmissions per case were higher for the imported cases than the locally transmitted cases (nine vs. three cases). The mean reporting delays were estimated to be 6.76 days (95% CI: 4.53, 9.28) and 2.57 days (95% CI: 1.57, 4.23) for imported and locally transmitted cases, respectively. The mean incubation period was estimated to be 5.53 days (95% CI: 3.98, 8.09) and was shorter than the mean serial interval of 6.45 days (95% CI: 4.32, 9.65). The $R_{0}$ was estimated to be 0.40 (95% CI: 0.16, 0.99), accounting for the local and imported cases. The fewer secondary cases and shorter reporting delays for the locally transmitted cases suggest that contact tracing of imported cases was effective at reducing further transmissions, which helped to keep $R_{0}$ below one and the overall transmissions small.

## 1. Introduction

Since December 2019, there has been an outbreak of pneumonia of unknown origin in Wuhan, Hubei, China. The causative agent was identified as severe acute respiratory syndrome coronavirus 2 (SARS-CoV-2), as defined by the World Health Organization (WHO) [1]. The associated disease, COVID-19 [2], has shown to cause flu-like symptoms such as fever, dry cough, dyspnea, and fatigue [3,4]. Although wild animals, e.g., bats [5], are suspected to be the source of infection, human-to-human transmissions mainly accelerated new infections in China, which may have been influenced by the massive human migrations during the Chinese New Year (Chun Yun) [6]. Afterward, the virus spread globally and WHO classified COVID-19 as a pandemic on 11 March 2020 [7]. As of 10 January 2021, the USA (21.7 million cases), India (10.4 million cases), and Brazil (8.0 million cases) were the top three countries by cumulative cases [8].

Studies on the initial spread of the disease in the Wuhan area provided useful insights on the epidemiological characteristics of COVID-19. An important characteristic is the basic reproduction number (*R*_0_), which represents the average number of secondary cases using an index case. Most of the previous studies on COVID-19 reported *R*_0_ to be 2–3 [6,9,10] although larger estimates (e.g., around six) were also reported in other studies [11,12]. Other key epidemiological parameters, such as incubation period (4–7 days) and serial interval (5–19 days), were found to be similar to those of other coronaviruses: Middle East respiratory syndrome coronavirus (MERS-CoV) and Severe acute respiratory syndrome coronavirus (SARS-CoV) [13].

The COVID-19 outbreak in Korea was initiated by importation from China (Appendix A) and local transmission remained limited until a superspreading event (SSE) occurred in a religious community early February [14]. This study was motivated by observing the small outbreak of COVID-19 in Republic of Korea in the early phase of disease spread. It can be interpreted that the control interventions were effective at preventing new infections. However, statistical estimation of the epidemiological factors has not been used to analyze the small outbreak in Korea. In this study, we provide a statistical characterization of the small outbreak by analyzing the individual data of the first 28 confirmed cases reported from 20 January to 10 February 2020. 

## 2. Materials and Methods

We sought to illustrate some key epidemiological variables of COVID-19 transmission in Korea and compared them with estimates from other countries.

### 2.1. Epidemiological Data

Based on the official reports from the Korea Disease Control and Prevention Agency (KDCA) [14] and the previous study [15], we collated individual data of the first 28 cases. The data included the dates of symptom onset ($t_{o}$), confirmation ($t_{c}$), exposure ($t_{e}$), discharge from the hospital ($t_{d}$), entry to Republic of Korea ($t_{in}$), and the infector ID, which represents the infector–infectee relationship. All datasets analyzed in this study are summarized in Appendix A. Figure 1A illustrates the progression of the COVID-19 epidemic based on the dates of confirmation. The information on who infected whom over the course of the outbreak and infector–infectee pairs appears in Figure 1B. Of the 18 cases for which we have complete information on the transmission history, the number of imported, primary, and secondary cases were 6, 9, and 3, respectively.

### 2.2. Statistical Inference

Using the maximum likelihood method, we estimated the following key epidemiological variables: (i) *P*_1_: reporting delay of imported cases between the symptom onset and the confirmation (i.e., time delay $d_{1}=t_{c}-t_{o}$); (ii) *P*_2_: reporting delay of local cases between the symptom onset and the confirmation (i.e., $d_{2}=t_{c}-t_{o}$); (iii) *P*_3_: time between the confirmation and discharge from the hospital (i.e., $d_{3}=t_{d}-t_{c}$), excluding a case (ID 9) who died from COVID-19; (iv) *P*_4_: time between the symptom onset and discharge from the hospital (i.e., $d_{4}=t_{d}-t_{o}$), where a case (ID 9) who died from COVID-19 was also excluded; (v) *P*_5_: incubation period (i.e., time between the exposure and the symptom onset, $d_{5}=t_{o}-t_{e}$); and (vi) *P*_6_: serial interval (i.e., time between symptom onsets of infector–infectee pairs, $d_{6}=t_{o}^{infectee}-t_{o}^{infector}$), where $t_{o}^{infectee}$ and $t_{o}^{infectee}$ represent the dates of symptom onset of infectee and infector, respectively. To account for the data that were reported daily, the continuous probability density function, f(t,θ), was defined at time *t*. Here, the parameter θ represents a vector of the mean (μ) and standard deviation (σ) of the probability distribution, i.e., θ=(μ,σ). The likelihood function for each time delay *P*_k_ is defined as:$$L\left(\theta ;d_{k}\right)=\overset{m_{k}}{\underset{i=1}{\prod }}f\left(d_{k}\left(i\right);\theta \right),$$
where *m*_k_ is the total number of cases in time delays, *P*_k_, and $d_{k}$ is the vector of time delay of the corresponding period, *P*_k_. To estimate the periods *P*_1_*–P*_6_, we employed three probability distributions that are commonly used in epidemiology: gamma, log-normal, and Weibull distributions [16,17]. W additionally analyzed the periods *P*_1_*–P*_6_ using four other distributions shown in Appendix A.

We compared the performance of each statistical model by calculating the second-order Akaike information criterion (AICc) and Bayesian information criterion (BIC). To compute the 95% confidence interval (95% CI), parametric bootstrap samples were generated from the multivariate normal distribution of the variance–covariance matrix, which was obtained from the Hessian matrix for estimated values. The 95% CI was calculated at the 2.5th and 97.5th percentile values of the resampled distribution. Among the commonly used three statistical models, the best fitting distributions were chosen by the minimum AICc values for the epidemiological periods *P*_1_*–P*_6_, separately. 

### 2.3. Transmission Model

The *R*_0_ was estimated using the renewal equation used in the previous studies [18,19,20]:(1)$$E(c_{t})=R_{0}\overset{t}{\underset{\tau =0}{\sum }}\left(c_{t-\tau }+\alpha j_{t-\tau }\right)f\left(\tau ;\theta \right),0\leq \alpha \leq 1.$$ Here, $c_{t}$ is the daily number of local cases, $j_{t}$ is the daily number of imported cases, and *E*(.) represents the expected value calculated from the right-hand side of the renewal Equation (1). The parameter α represents the relative transmissibility of imported cases to locally transmitted cases. If α=0, there would be no secondary cases caused by the imported cases. The 95% CI was computed from the parametric bootstrapping with 1000 samples of the mean and standard deviation of the serial interval distribution. We assumed that the COVID-19 cases, $c_{t}$, follow a Poisson distribution, an approach adopted in previous studies [21,22,23], which leads to the likelihood function with unknown parameter $R_{0}$ as follows:$$L\left(R_{0};c_{t}\right)=\overset{t_{n}}{\underset{t=1}{\prod }}\frac{E{\left(c_{t}\right)}^{c_{t}}exp\left(-E\left(c_{t}\right)\right)}{c_{t}!},$$
where $t_{n}$ is the final time of symptom onset.

### 2.4. Ethical Considerations

We used the data available in Appendix A. The datasets were already fully anonymized and did not include any personally identifiable information. Thus, ethical approval was not required for this analysis.

## 3. Results

### 3.1. Estimation of Epidemiological Periods

The epidemiological periods were estimated using three different probability distributions (gamma, log-normal, and Weibull) that are commonly used for modeling epidemiological periods. The results are shown in Appendix A. Figure 2 illustrates comparisons of observed periods based on the best-fitting distributions supported by the minimum value of AIC_C_ among three distributions. The corresponding parameter estimates are described in Table 1.

Appendix A describes the results of fitting seven different probability distributions. We found even lower AIC_C_ values than the best-fitting distributions described in Figure 2, though differences in their AIC_C_ values were very small. For the reporting delay in imported cases (*P*_1_), the Weibull distribution provided the lowest AICc with the estimated mean of 6.76 days (95% CI: 4.53, 9.28) and standard deviation (SD) of 4.74 days (95% CI: 3.05, 8.70). In addition, the mean reporting delay for the locally transmitted cases (*P*_2_) was estimated to be 2.57 days (95% CI: 1.57, 4.23). This implies that the imported cases were likely to generate more secondary transmissions than the locally transmitted cases. Second, the period between confirmation and discharge (*P*_3_) and the period between symptom onset and discharge (*P*_4_) were estimated to be 15.91 days (95% CI: 14.06, 17.72) and 21.87 days (95% CI: 19.97, 23.72), respectively. The time between symptom onset and discharge was calculated to be about 2 weeks, providing some information on the natural history of infection [15]. Third, the mean incubation period (*P*_5_) was estimated to be 5.53 days (95% CI: 3.98, 8.09) using a log-normal distribution. This value is similar to the estimate in a previous study [13] (5.2 days; early stages of the outbreak in Wuhan, China), and values reported by Lauer et al. [24] (5.1 days; a large sample of 181 cases in China). Lastly, the mean serial interval (*P*_6_) was also estimated to be 6.45 days (95% CI: 4.32, 9.65), which is similar to that observed during the early stages of the outbreak in Wuhan, China (7.5 days [13] and 6.3 days [25]). Additionally, the reporting delay of imported cases between entry to confirmation was estimated with a mean of 8.39 days (95% CI: 6.09, 10.80) and SD of 4.63 days (95% CI: 3.09, 7.94) from the Weibull distribution, shown in Appendix A. The reporting delay for the locally transmitted cases (*P*_2_), estimated to be 2.57 days, was shorter than the reporting delay in imported cases between the entry to confirmation, estimated to be 6.76 days. This may reflect that increased surveillance and case isolation would have limited later-stage transmissions over the course of the infection.

### 3.2. Basic Reproduction Number

To summarize, the longer reporting delay for imported cases compared to the locally transmitted cases hints that imported cases played a dominant role in the early stages of the outbreak in Korea. This is supported by the generation-specific reproduction numbers ($R_{0}=0.48,R_{1}=0.56,R_{2}=0.33$) [15], where $R_{n}$ represents the reproduction number for generation *n*. In other words, the basic reproduction number of the imported cases ($R_{1}$) was higher than that of the locally transmitted cases ($R_{2}$). According to the transmission rates from imported cases (α), $R_{0}$ was estimated to be 0.40 (95% CI: 0.16, 0.99) if imported and locally transmitted cases are equally transmissible (i.e., α=1) over the dates of symptom onset. If imported cases are assumed to be less transmissible than the locally transmitted cases (e.g., because of quarantine at the airport), estimates for the $R_{0}$ increase accordingly (Table 2).

Similar values were also achieved on other occasions, for example, in the MERS-CoV outbreak, but these are lower than those of SARS-CoV in China [10]. However, the incubation period and serial interval were in line with the early stages of the SARS-CoV outbreak in China. Moreover, the estimated serial interval was approximately 2.7 days longer than the estimated incubation period [15]. Nevertheless, there was no large change in the transmission trends, i.e., the serial interval is longer than the incubation period. Our results imply that a pre-symptomatic transmission accompanied with a huge number of infections was otherwise not likely during the early stages of COVID-19 in Korea.

## 4. Discussion

We analyzed the data of 28 COVID-19 cases confirmed during 20 January–10 February to describe the dynamics of subcritical transmissions during the early stages of COVID-19 transmissions in Korea. Cases that occurred following 18 February were driven mainly by superspreading events in religious communities; therefore, transmission dynamics were quite different compared to the earlier period. In addition, data collected during superspreading events were not as reliable as those collected during the earlier period. While it does not provide a complete picture of COVID-19 transmission in Korea, the present study, we think, provides the most extensive analyses of transmissions during the early stages. In the present study, we characterized the epidemiology of the limited local transmissions during the early stages of the COVID-19 outbreak in Korea. This was achieved by estimating the incubation periods, serial intervals, reporting delays, and the reproduction number of the first 28 confirmed COVID-19 cases in Korea. Two main insights emerged from our analyses. First, the imported cases played a dominant role in generating transmissions, while the overall basic reproduction number accounting for the imported and locally transmitted cases was estimated at 0.40 (95% CI: 0.16, 0.99). Moreover, the delays from the symptom onset to confirmation were longer for imported cases than the locally transmitted cases; this difference in delays is partly responsible for the difference between $R_{1}$ and $R_{2}$, shown in a previous study [15] (i.e., $R_{1}=0.56,R_{2}=0.33$). Second, the serial interval was longer than the incubation period (i.e., $d_{6}>d_{5}$), which suggests pre-symptomatic transmissions were not frequent [26].

As of 11 October 2020, COVID-19 spread to all continents [7]. There is active and ongoing research on the effectiveness of disease prevention policies, as in our work. However, not every infection is detected by syndromic surveillance [27,28]. This is evidenced in our patient dataset, where most of the imported cases were confirmed upon arrival in Korea. In other words, the actual times of infections or symptom onsets are likely to be earlier than the confirmations at the airport. The transmission tree exhibited that the difference in the confirmation dates for some successive infector–infectee pairs was very short; for example, a family (ID 25, ID 26, and ID 27 in Figure 1) was confirmed on the same day. This is unrealistic considering the incubation period.

Our analyses showed that the incubation period (*P*_5_) is shorter than the serial interval (*P*_6_), with their estimates being 5.53 days (95% CI: 3.98, 8.09) and 6.45 days (95% CI: 4.32, 9.65), respectively. This relationship did not hold for other settings and the difference may reflect the extent of intervention programs that reduced pre-symptomatic transmissions. In Nishiura et al. [26], who analyzed the data from Germany, *P*_5_ and *P*_6_ were 5.2 and 4.0 (or 4.6) days, respectively. From the Singapore outbreak, *P*_5_ and *P*_6_ were estimated to be 5.99 and 4.0 days, respectively [13]. Tindale et al. [29] reported *P*_5_ = 8.68 days and *P*_6_ = 5.0 days for Tianjin, China, and Yang et al. [30] reported *P*_5_ = 6.0 days and *P*_6_ = 4.6 days for Hubei, China.

Combining those two pieces of analysis, the risk of pre-symptomatic transmission was low in the Republic of Korea during the early stages, since non-pharmaceutical interventions such as social distancing, wearing masks, and contact tracing were operating well. Thus, it resulted in a relatively small basic reproduction number (*R*_0_ = 0.68–1.77) compared to the other outbreaks (SARS-CoV and SARS-CoV-2), though a similar level was reported regarding MERS-CoV (Table 3).

Our study has several limitations, such as not including cases confirmed after 18 February 2020, but focusing on the early period of transmission where contact tracing for the confirmed cases was totally identified. This study may serve as a baseline for future studies on control intervention for COVID-19. It is clear that the initial spread of COVID-19 in Korea was well-controlled by effective contact tracing and non-pharmaceutical interventions, although the imported cases impacted the spread of the outbreak of COVID-19. In addition, the KDCA and local authorities had close to full control of public health surveillance (e.g., epidemic investigations). In this context, non-pharmaceutical interventions, especially contact tracing, would have been more effective than public health policies such as social distancing [41]. Regarding other countries, one may consult a number of sources [27,28,31] for discussions on the impact of potential intervention strategies including travel restrictions without using the officially reported data.

Our renewal equation model did not consider the asymptomatic and pre-symptomatic transmission of COVID-19. There was just one asymptomatic case (ID 18) in our data set and pre-symptomatic transmissions did not seem to be common based on the serial interval being longer than the incubation period. However, such modes of transmission may become highly relevant and may eventually contribute to formations of clusters that are accompanied with large infections, such as SSE. For these reasons, the results presented in other papers should be considered alongside real-time data when making serious decisions such as public health policy [41,42,43].

Despite several limitations, the present analysis is one of few studies available on the early transmission of COVID-19 in the Republic of Korea. From the statistical models developed in this paper, we deduced that the early outbreaks initiated by imported cases were effectively halted by non-pharmaceutical interventions such as wearing masks and social distancing. We think that such modelling not only explains the trends in an epidemic outbreak, but also enrich the interpretation of possible causes, which is of great merit to society.

## 5. Conclusions

Limited transmissions of COVID-19 during the early stages of COVID-19 in Korea can be explained through the following two observations: First, reporting delays for the local cases were shorter than for the imported cases, which indicates that further transmissions were effectively prevented (i.e., low *R*_0_). Second, pre-symptomatic transmission seemed to be rare during this period, as shown in that the incubation period was shorter than the serial interval. 

## Figures and Tables

**Figure 1 ijerph-18-01265-f001:**
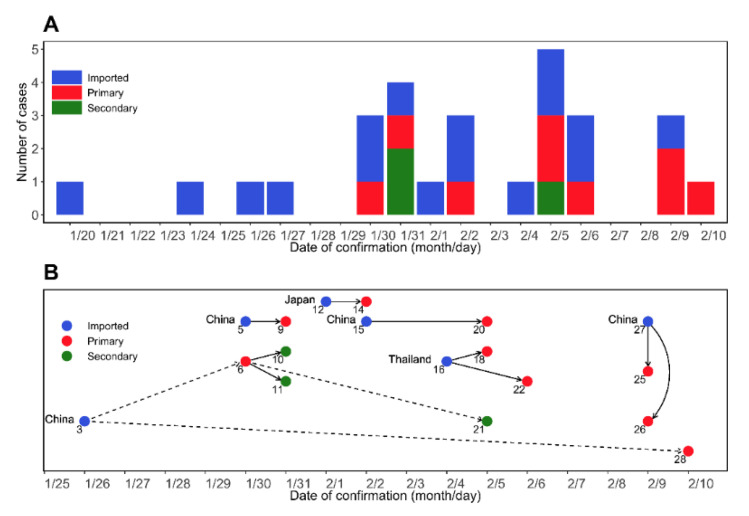
COVID-19 epidemic curve and chain of contagion in a sample of patients in the Republic of Korea. 20 January–10 February 2020. (**A**) An epidemic curve over the confirmed dates. The confirmed cases were grouped as imported cases (blue), primary (red), and secondary (green) infections. Note that the imported cases are the patients who were already infected. (**B**) A chain of COVID-19 infection by confirmation date in Korea from 20 January to 10 February 2020. The imported cases (blue) are annotated with the country of importation. Solid arrows and dashed arrows represent transmission relationships that are between family members and friends, respectively.

**Figure 2 ijerph-18-01265-f002:**
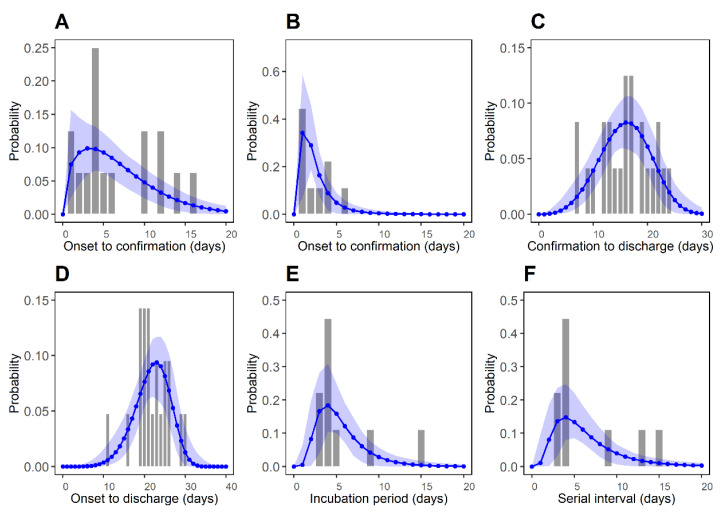
Best-fitted distributions for epidemiological periods *P*_1_–*P*_6_. Bars represent the observed data and blue dots represent the estimated values, respectively. Blue shaded regions represent the 95% CI from 1000 samples. (**A**) Reporting delay of imported cases (*P*_1_: symptom onset to confirmation). (**B**) Reporting delay for local cases (*P*_2_: symptom onset to confirmation). (**C**) Time between the confirmation and the discharge from the hospital (*P*_3_: confirmation to discharge). (**D**) Time between the symptom onset and the discharge from the hospital (*P*_4_: symptom onset to discharge). (**E**) Incubation period (*P*_5_). (**F**) Serial interval (*P*_6_).

**Table 1 ijerph-18-01265-t001:** Parameter estimates for the epidemiological periods (*P*_1_–*P*_6_).

Time Delay	Distribution	*m*	Mean (day)	SD* (day)	AIC_C_	BIC
Symptom onset to confirmation for imported cases (*P*_1_)	Weibull	16	6.76 (4.53, 9.28)	4.74 (3.05, 8.70)	95.11	95.73
Symptom onset to confirmation for local cases (*P*_2_)	Log-normal	9	2.57 (1.57, 4.23)	1.99 (0.72, 4.97)	37.51	35.90
Confirmation to discharge (*P*_3_)	Weibull	24	15.91 (14.06, 17.72)	4.66 (3.63, 6.30)	147.10	148.88
Symptom onset to discharge (*P*_4_)	Weibull	21	21.87 (19.97, 23.72)	4.29 (3.42, 6.02)	124.30	125.72
Incubation period (*P*_5_)	Log-normal	9	5.53 (3.98, 8.09)	2.96 (1.28, 6.09)	47.65	46.04
Serial interval (*P*_6_)	Log-normal	9	6.45 (4.32, 9.65)	4.16 (1.67, 8.87)	52.45	50.85

*m,* the number of data in a dataset. SD*, standard deviation, where 95% CI is shown in parenthesis. In-sample errors were computed by the second order Akaike information criterion (AIC_C_) values and the Bayesian information criterion (BIC) for three different distributions (gamma, log-normal, and Weibull). AIC_C_ and BIC are defined by ${\mathrm{AIC}}_{C}=2n-2log\left(L\right)+\frac{2n^{2}+2n}{m-n-1}$ and BIC=−2log(L)+log(m)n, where *n* represents the number of parameters and *L* is the maximized likelihood of a fitted delay function.

**Table 2 ijerph-18-01265-t002:** Estimates for the basic reproduction number varying with effects of imported cases.

α	$R_{0}$	95% CI*
0.0	1.27	(0.48, 2.95)
0.1	1.05	(0.40, 2.47)
0.3	0.78	(0.30, 1.86)
0.5	0.62	(0.24, 1.49)
0.7	0.51	(0.20, 1.24)
1.0	0.40	(0.16, 0.99)

95% CI*: Profile-likelihood-based 95% confidence intervals.

**Table 3 ijerph-18-01265-t003:** Summary of the basic reproduction number ($R_{0}$ ), serial interval, and incubation period for three different coronavirus outbreaks.

Virus	Epidemics	*R* _0_	Incubation Period	Serial Interval
**SARS-CoV**	**China (Beijing)** **2003**	1.88 (mean) overall [31]	5.7 (SD 9.7) [32]	NA
	**Hong Kong** **2003**	1.70 (95% CI: 0.44, 2.29) overall [33], 2.7 (95% CI: 2.2, 3.7) in the early phase (excluding SSE) [34] 0.14–1 in the later phase (excluding SSE) [34]	4.6 (95% CI: 3.8, 5.8) [35]	(8, 12) [36]
**MERS-CoV**	**Saudi Arabia** **2013–2014**	0.45 (95% CI: 0.33, 0.58) overall [37]	5.0 (95% CrI: 4.0, 6.6) [38]	6.8 (SD 4.1) [37]
	**Republic of Korea** **2015**	0.91 (95% CI: 0.36, 1.44) overall, 2.0–8.1 in early phase (including SSE) [39]	6.9 (95% CrI: 6.3, 7.5) [38]	12.4 (SD 2.8) [40]
**SARS-CoV-2**	**China** **2019–2020**	2.58 (95% CrI: 2.47, 2.86) [6]	5.2 (95% CI: 4.1, 7.0) [13]	7.5 (95% CI: 5.3, 19) [13]
	**Republic of** **Korea2020**	0.48 (95% CI: 0.25, 0.84) in early phase (before SSE) [15]	3.9 (0–15) [15]	6.6 (3–15) [15]

SD: Standard deviation; NA: not available; SSE: superspreading event; 95% CI: 95% confidence interval; 95% CrI: 95% credible interval.

## Data Availability

The epidemiological data for the 28 cases of COVID-19 is available in Appendix A.

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
