# Peer review of "Subcritical Transmission in the Early Stage of COVID-19 in Korea"

_ijerph, 2021, doi:10.3390/ijerph18031265_

Round 1

Reviewer 1 Report

This work is entitled to publicatrion and I do recommend this.

I have very slight comments on form:

  1. in line 196: Instead of: "...between ?1 and ?2⁡can be explained partly by that delays from the illness onset to 196 confirmation were longer for imported cases than the locally..." A clearer format might be adopted, to make for easier reading, something like: "The differencdelays from the illness onset to 196 confirmation were longer for imported cases than the locally transmitted cases can partly explain the difference between ?1 and ?2."; 
  2. In line 221, if authors accept a suggestion, I would indicate, instead of: "Our study has several limitations. We did not...", another form:  "Our study does have limitations, such as not including cases confirmed after February 18, 2020 but focused..."
  3. Likewise, in line 234, instead of: "...set and pre-asymptomatic transmissions seemed to be not common based on that serial..." use of: "...set and pre-asymptomatic transmissions did not seem to be common based on that serial...";
  4. In line 239, a suggestion: "Although there are several limitations, the present analysis...", maybe: In spite of several limitations, the present analysis..."

As noted above, these are but suggestions. And with this I must repeat thet this work merits publication. 

Author Response

We are grateful for the valuable comments by the reviewers and the opportunity to revise our manuscript. We addressed all of the reviewer's comments in the attached file and have made an effort to better explain the relevance of our results in the revised manuscript.

Reviewer 2 Report

In this paper, the authors collect data from official reports of the Korea Disease Control and Prevention Agency and from a study done previously, of the first 28 cases of COVID-19 reported in South Korea. They show that the first reported cases were imported from China, Japan and Thailand and they track primary and secondary infections already in Korean territory. They use the maximum likelihood method to estimate some variables such as the delay time between the onset of the disease and confirmation, both for imported and local cases, the delay in time between confirmation and discharge from the hospital, incubation time, among others (six variables in total).
With these variables, they establish the likelihood function of the delay time on the 28 cases analyzed. They propose the statistical models and the estimated values for the given confidence intervals and choose the model that gives the best results for the six variables of time.
Estimating the basic reproductive number using a method reported in a previous work, they propose a transmission model.

Suggestions:

1. Figure 2 is not clear, under each graph there is a legend that, in some cases, does not correspond to the explanation given in the figure caption.
2. Reference 14 is not complete.
3. Accept the paper after correcting.

Author Response

(The authors gave the same response as above.)

Reviewer 3 Report

This study reported a statistical characterization of COVID-19 transmissions during the early stages of the outbreak in Korea. Some parameters including the reporting delay, incubation period, ?0, were analyzed. This is a short manuscript, and there are some major concerns.

Please explain the reason in detail why only choosing the COVID-19 cases from January 26 to February 10 2020?

Only 28 cases were involved in this study for analysis. For a so-small number, it is hard to take a solid conclusion. Especially for the estimated the basic reproduction number (?0) in this study is significantly lower than previous studies, why?

Author Response

(The authors gave the same response as above.)

Reviewer 4 Report

No comments

Author Response

(The authors gave the same response as above.)

Reviewer 5 Report

This manuscript describes estimation of the distribution of several epidemic parameters using data from early transmission of SARS-CoV-2 in South Korea. The methodology is fine, there is just not anything particularly novel in the manuscript.

  • The authors need to better describe how the renewal equation is being used to calculate R0. I don't see an R0 in any of the equations of the paper, so how did they find it?
  • If the authors used 7 different distributions for their analysis, why are they limiting the distributions in Figure 2 to Weibull and lognormal when other distributions provided a better fit?
  • Do the colors of the boxes in Tables S1 and S3 mean anything?
  • The colors are mislabeled in the caption of Fig. 1. It should be primary (red) and secondary (green)

Author Response

(The authors gave the same response as above.)

Round 2

Reviewer 3 Report

All my concerns are addressed.